# *Succinivibrionaceae* is dominant family in fecal microbiota of Behçet's Syndrome patients with uveitis

Duygu Tecer[1], Feride Gogus[1], Ayse Kalkanci[2]*, Merve Erdogan[2],
Murat Hasanreisoglu[3‡], Çagri Ergin[4‡], Tarkan Karakan[5‡], Ramazan Kozan[6‡],
Seda Coban[7‡], Kadir Serdar Diker[8]

1 Department of Physical Medicine and Rehabilitation, Division of Rheumatology, Gazi University Faculty of Medicine, Ankara, Turkey, 2 Department of Medical Microbiology, Gazi University Faculty of Medicine, Ankara, Turkey, 3 Department of Ophthalmology, Gazi University Faculty of Medicine, Ankara, Turkey, 4 Department of Medical Microbiology, Pamukkale University Faculty of Medicine, Denizli, Turkey, 5 Department of Internal Medicine, Division of Gastroenterology, Gazi University Faculty of Medicine, Ankara, Turkey, 6 Department of General Surgery, Gazi University Faculty of Medicine, Ankara, Turkey, 7 Medical student, Gazi University Faculty of Medicine, Ankara, Turkey, 8 Department of Microbiology, Faculty of Veterinary Medicine, Adnan Menderes University, Aydin, Turkey

☉ These authors contributed equally to this work.
‡ These authors also contributed equally to this work.
* aysekalkanci@email.com, kalkanci@gazi.edu.tr

**Data Availability Statement:** Nucleotide sequences of 1797 taxa were deposited in NCBI GenBank with accession numbers MT573971 to MT575767 (file SUB7562864).

## Abstract

Behçet's Syndrome (BS) is a multisystem vasculitis with various clinical manifestations. Pathogenesis is unclear, but studies have shown genetic factors, innate immunity and auto-inflammation to have an important role in the disease course. Diversity in the microbial community of gut microbiota may significantly contribute to the activation of the innate immune system. The clinical features of BS present themselves in clusters and each cluster may be a consequence of different disease mechanisms. For this reason we aimed to investigate the gut microbiota of BS patients with uveitis. In addition to healthy controls, we have aimed to compare the gut microbiota of BS with that of Familial Mediterranean Fever (FMF) and Crohn's Disease (CD) as both diseases have innate and autoinflammatory features in their pathogenesis. Seven patients with BS, 12 patients with FMF, 9 patients with CD and 16 healthy controls (HC) were included in the study. Total genomic DNAs were isolated from fecal samples of the patients. Partial 16S rRNA gene was sequenced using the PGM Ion Torrent (Thermo Fisher Scientific, Waltham, MA, USA) for microbiota analysis. Statistical analysis showed that significant differences were detected on the microbial community of four groups. *Succinivibrionaceae* is dominant and the signature family, whereas Bacteroides was absent in BS patients.

## Introduction

Behçet's Syndrome (BS) is a chronic, multisystemic, inflammatory disease of unknown origin. Clinical and immunological characteristics of BS may suggest the possibility of more than one

**Funding:** This project was supported and funded with a grant of Turkish League Against Rheumatism. The funder had no role in study design, data collection and analysis, decision to publish, or preparation of the manuscript.

**Competing interests:** The authors have declared that no competing interests exist.

mechanism in the pathogenesis and, some clinical features of BS show distinctive clusters [1]. Furthermore, although carrying HLA-B51 increases the risk of developing BS, it does not seem to have an effect on the prognosis except for ocular involvement [2–4].

The recurrent, self-limited clinical features resemble the autoinflammatory syndromes, like Familial Mediterranean Fever (FMF) [1]. Although BS cannot be definitely classified as an autoinflammatory disease, it shares some clinical and pathophysiologic features with autoinflammatory diseases like a relapsing-remitting disease course, neutrophil hyperactivity, enhanced inflammatory response with the overexpression of pro-inflammatory cytokines such as IL-1, absence of specific autoantibodies and antigen-specific T cells [5–7]. FMF, a prototype autoinflammatory disorder was chosen as one positive control. Most of the BS's disease manifestations can be seen in patients with FMF and both diseases are more prevalent in the Mediterranean basin and the Middle East [1]. Moreover, compared to healthy individuals MEFV gene mutations are common in patients with BS which may have a role in the pathogenesis of BS [8, 9].

The other diseased control group was Crohn's disease (CD) which is a type of inflammatory bowel disease that may affect any part of the gastrointestinal tract. BS shares many characteristics with CD including genetic background, clinical manifestations, endoscopic and histologic feature and also therapeutic strategies [10]. IL-10 and IL23R variants were observed in both diseases, which suggest that these two diseases may have similar genetic backgrounds and pathogenesis [11–13]. Recently genome-wide association studies (GWAS) identified newly susceptibility loci shared between BS and CD, which include FUT2, ADO-EGR2, RIPK2, LACC1, and IRF8 [14]. Innate and adaptive immunity are involved in the pathogenesis of both diseases. Many of the extra-intestinal manifestations of CD, such as uveitis, arthritis, pyoderma gangrenosum, erythema nodosum, oral ulcers, overlap with the manifestations of BS [10].

Previous studies of BS have shown an increase in the innate immune response which may suggest the possibility of an infectious trigger [15–18]. Isolation of *Prevotella* and *Staphylococcus* from pustular lesions and an increased colonisation of the saliva with *Streptococcus mutans* further enhance this possibility [19, 20]. The several species of cross-reactive Streptococci antigens, *Haemophilus influenzae*, herpes simplex virus-1 and various fungal species have been investigated as potential etiological triggers.

The gut microbiota is an active component of the immune system. It plays an important role in the formation of the immune system in early life and in the continuation of immunohomeostasis throughout life. Dysbiosis, the imbalance in the gut microbiota, may lead to many serious metabolic and inflammatory pathologies [21].

The immunopathogenesis of BS involves a possible altered peptide presentation triggered by an altered microbiome by human leucocyte antigen (HLA)-B51 of antigen presenting cells (APCs) [4]. Hence, the microbiota analysis of the BS patients is worthier to investigate. BS can involve the skin, mucosa, joints, eyes, vessel, nervous and gastrointestinal system. The clinical course seems to follow four major clusters: 1) oral and genital ulcers, erythema nodosum 2) superficial and deep vein thrombosis 3) uveitis 4) papulopustuler skin lesions and joint involvement [2]. Due to this heterogeneous spectrum, BS can hardly be considered as a uniform disease, moreover different pathogenic mechanisms may be present in each cluster [22, 23].

The present study aims to investigate the gut microbiota of patients with BS. As uveitis is a clinical cluster on its own and HLA-B51 has an effect on its prognosis, we have selected a subgroup of BS patients with the uveitis involvement only. In addition to the healthy controls, we have aimed to compare the gut microbiota of BS with that of FMF and CD as positive control groups where like BS, the innate immune response has a major role [24].

## Materials and methods

### Patients

Participants were recruited from Gazi University Faculty of Medicine, Department of Physical Medicine and Rehabilitation, Division of Rheumatology, Ankara, Turkey between March 2016-March 2017. A total of 44 participants (7 patients with BS with uveitis, 12 patients with FMF, 9 patients with CD and 16 healthy controls (HC) were included in this case-control study. Informed written consent was obtained from each participant, according to the principles of Helsinki Declaration. Approval for the study was obtained from the Committee on Human Research Ethics of Zekai Tahir Burak Women's Health Education and Research Hospital (dated: 23/02/2016, decision number: 11/2016). The diagnosis of BS was based on the diagnostic criteria of the International Study Group of Behcet's disease [25]. The Tel-Hashomer criteria were used to make the diagnosis of FMF [26]. The diagnosis of CD was based on the clinical evaluation including a detailed history, physical examination, and combination of endoscopic findings, histology, radiologic findings and laboratory investigations. Exclusion criteria applied to the participants were as follows: recent (<6 months prior to the sample collection) treatment with probiotics and antibiotics; history of concomitant diseases, such as autoimmune disease, infections, malignancies, gout, obesity and diabetes, history of gastrointestinal surgery such as gastrectomy, bariatric surgery or colectomy. A written informed consent was obtained from all participants according to the principles of the Helsinki Declaration. Demographic and clinical characteristics were recorded. Healthy controls; 6 males (39-50y) and 10 females (24-52y) were matched for age and sex. Demographical and clinical characteristics of participants at the time of fecal sample collection were listed in Table 1.

### Stool samples and genomic DNA extraction

The fecal samples of patients with BS were obtained during an active uveitis attack. The fresh stool samples were transported to the Gazi University, Faculty of Medicine, Department of Medical Microbiology Laboratory immediately for analysis. Stool samples were preserved in -80˚C for molecular analysis. Genomic DNA was extracted using the QIAamp® DNA Stool Kit (QIAGEN Inc., CA, USA) from each fecal sample. Minor revision at the elution step was performed by using a 30 μl elution buffer. The extracted DNA was quantified using spectrophotometer at wavelengths of 260 nm and stored at -20˚C until PCR amplification.

### 16S rRNA gene amplification

DNA samples were quantified and equalized using the Qubit 2.0 System. Partial 16S rRNA gene sequences were amplified from extracted DNA using the 16S Metagenomics Kit (Life Technologies, Italy). Hypervariable V2-4-8 regions and V3-6, 7–9 regions were amplified by two primer sets. A total of 1,5–3 ng/μl DNA was subjected in this PCR step. PCR conditions were as follows: 95˚C for 10 min, 25 cycles of 95˚C per 30s denaturation; 58˚C per 30s annealing; and 72˚C 20s extention; followed by 72˚C per 7 min. Amplicons were checked by electrophoresis. Further steps were followed after the visualization of PCR products on 2% agarose gel.

### Next generation sequencing

Partial 16S rRNA gene sequenced using the PGM Ion Torrent (Thermo Fisher Scientific, Waltham, MA, USA). AMPure XP DNA purification beads and Invitrogen DynaMaq magnet apparatus were used for the purification of amplicons. DNA concentration of the PCR product was equalized as 40 ng/μl through the Qubit 2.0 System (Qubit ds DNA HS Assay kit). Finally,

**Table 1. Demographical and clinical characteristics of participants at the time of fecal sample collection.**

| | Behçet's Syndrome (n:7) | Crohn's disease (n:9) | Familial Mediterranean Fever (n:12) | Healthy Controls (n:16) |
|---|---|---|---|---|
| **Age, years** | 35.57±6.60 | 35.00±5.27 | 32.17 ± 8.64 | 39.38±7.69 |
| **Men/women** | 5/2 | 3/6 | 6/6 | 6/10 |
| **Disease characteristics** | | | | |
| Oral aphthosis, n (%) | 5 (71.43%) | 1 (11.11%) | 0 | 0 |
| Genital ulcers, n (%) | 1 (14.29%) | 0 | 0 | 0 |
| Skin lessions, n (%) | 3 (42.86%) | 1 (11.11%) | 0 | 0 |
| Uveitis, n (%) | 7 (100%) | 0 | 0 | 0 |
| Gastrointestinal system involvement, n (%) | 0 | 9 (100%) | 0 | 0 |
| Central nervous system involvement, n (%) | 1 (14.29%) | 0 | 0 | 0 |
| Vascular involvement, n (%) | 0 | 0 | 0 | 0 |
| Arthritis, % | 0 | 2 (22.22%) | 2 (16.67) | 0 |
| Pathergy positivity | 2 (28.57%) | NA | NA | NA |
| HLA-51 | 4 (57.14%) | NA | NA | NA |
| **Medications** | | | | |
| Colchicine, n (%) | 6 | 0 | 12 (100%) | 0 |
| Steroid, n (%) | 0 | 0 | 0 | 0 |
| 5-ASA/sulfasalazine, n (%) | 0 | 0 | 0 | 0 |
| Cyclosporine, n (%) | 1 | 0 | 0 | 0 |
| Azathioprine, n (%) | 4 | 9 | 0 | 0 |
| Methotrexate, n (%) | 0 | 0 | 0 | 0 |
| Infliximab, n (%) | 0 | 0 | 0 | 0 |

NA, not applicable

the reactions were combined in equimolar concentrations to create a mixture composed by 16S rRNA gene amplified fragments of each sample. This composite sample was used for library preparation. The libraries were created by using the Ion Plus Fragment Library Kit (Life Technologies). Barcodes were also adapted to each sample using the Ion Express Barcode Adapters 1–16 kit (Life Technologies). Agencourt AMPure XP Reagent and DynaMaq was used for the purification of barcoded products. Emulsion PCR was carried out using the Ion OneTouch™ 2 System with the Ion PGM™ Template OT2 400 Kit Template (Thermo Fisher Scientific, Waltham, MA, USA). The sequencing was performed using Ion PGM™ Sequencing 400 on Ion PGM™ System using Ion 318™ Chip v2 with a maximum of 30 samples per microchip. Ion Torrent Personal Genome Machine System and Ion PGM Hi-Q kit were used according to supplier's instructions. Sequences were analyzed by Ion Reporter Software (Thermo Fisher Scientific, Waltham, MA, USA). Software identified the microorganisms at phylum, class, order, family, genus and species level.

### Sequence processing and statistical analysis

Obtained sequences were transferred into Operational Taxonomic Units (OTUs). A similarity of 99% was accepted as the same phylotype. Relative abundance data were individually weighted for each taxon level. Observations with a weighted average relative abundance of less than 0.05% at phylum and rank, 0.05% at family rank, and 0.01% at genus rank were grouped to reducing variance.

Statistical analysis was performed using R (Ver 3.6.0, GPL-3 license). The relative abundances (expressed as parts per unit), calculated with *ade4* packet in R, were directly compared

in each taxonomic level (for different phylum, genus, species). The *vegan* package was used for alpha biodiversity. Alpha diversity is defined as the diversity within a community and is mainly measured by Chao 1 diversity indexes. Chao 1 index estimates richness in numbers of taxa frequencies. Within-sample microbial diversity was calculated as the Shannon diversity index based on the resulting profiles. All diversity and taxa graphs are presented by *ggplot2* R-package.

Based on relative abundance features, R packages *FactoMineR* and *factoextra* were used for dendrogram (Bray-Curtis similarity) and taxa distribution visualizations with groups while *ggfortify* and *ggplot2* were used for Principle Component Analysis (PCA).

To investigate the bacterial constitution in the gastrointestinal microbial communities from each group at each taxonomic level, taxa above the level of species were selected for following analysis based on the Kruskal-Wallis test results (by *dplyr R-package*). The topmost abundant taxa are presented in the form of heatmap (by *pheatmap R-package*). Pie graphs were presented by *plotly R*-package.

## Results

Results were analyzed by Curated MicroSEQ (R) 16S Reference Library v2013.1; Curated Greengenes v13.5 database. A total of 732959 raw reads were obtained from sequencing. After quality control steps, 150802 reads were ignored due to low number of copies ($<$10) and 384094 valid reads were obtained. The mapped reads in the sample was 230716. 16S rRNA community profiling of the samples yielded high sequencing depth, an average of 4,031 sequences/sample. Good's coverage was between 88% and 100% sequencing coverage at 99% similarity cutoff, indicating that the numbers of sequences were sufficient for the communities measured in all libraries. Nucleotide sequences of 1797 taxa were deposited in NCBI GenBank with accession numbers MT573971 to MT575767 (file SUB7562864).

Gut microbiota (GM) structure from 7 BS patients, 9 CD patients, 12 FMF patients and 16 HCs were characterized by the mean of sequencing of 16S rRNA gene. Groups were of similar age (p = 0.098). We first compared microbial diversity among the four study groups. Alpha diversity estimated statistically significant biodiversity differences between groups in terms of phylum, class, order, family and genus diversity (p$<$0.001). PCA analysis confirmed and showed that there were significant differences between the four groups as shown in Fig 1. Alpha diversity as defined by Chao 1 and Shannon diversity indexes are shown in Fig 2.

Fecal bacterial flora of BS consisted of Firmicutes (45%), Proteobacteria (23%) such as *Enterobacteriaceae* and *Prevotellaceae*. Fecal microbiota contained predominantly Bacteroidetes (53.2%) including *Bacteroides* and *Prevotella* generas in HC. Firmicutes such as *Bacillia*, *Clostridia*, followed consisting 31.2% of the bacterial community. FMF cases were found to be predominantly colonized by Firmicutes (39.3%), Bacteroidetes (32.2%) and Proteobacteria (13.6%). CD cases were colonized by *Enterobacteriacea* (43%), and other *Proteobacteria* groups, followed by Bacteroidetes (15.4%) and Firmicutes (8.8%).

Dominant classes were Clostridia, Gammaproteobacteria, Gammaproteobacteria, Negativicutes while dominant orders were Clostridiales, Bacteroidales, Selenomonadales, Aeromonadales in BS group.

The top five families in BS group were; *Succinivibrionaceae*, *Veillonellaceae*, *Prevotellaceae*, *Lachnospiraeae* and *Ruminococcaceae* while families in HC were; *Bacteroidaceae*, *Prevotellaceae*, *Ruminococcaceae*, *Lachnospiraceae*, and *Porphyromonadaceae* (Fig 3). *Bacteroidaceae* was at minor levels in CD and FMF groups while there was none in BS. *Enterobacteriaceae* seem to replace the flora in CD patients, followed by *Veionellaceae*. The top five families for each group were heat mapped in Fig 4. Other families were at minimum levels such as; *Bacteroidaceae*,

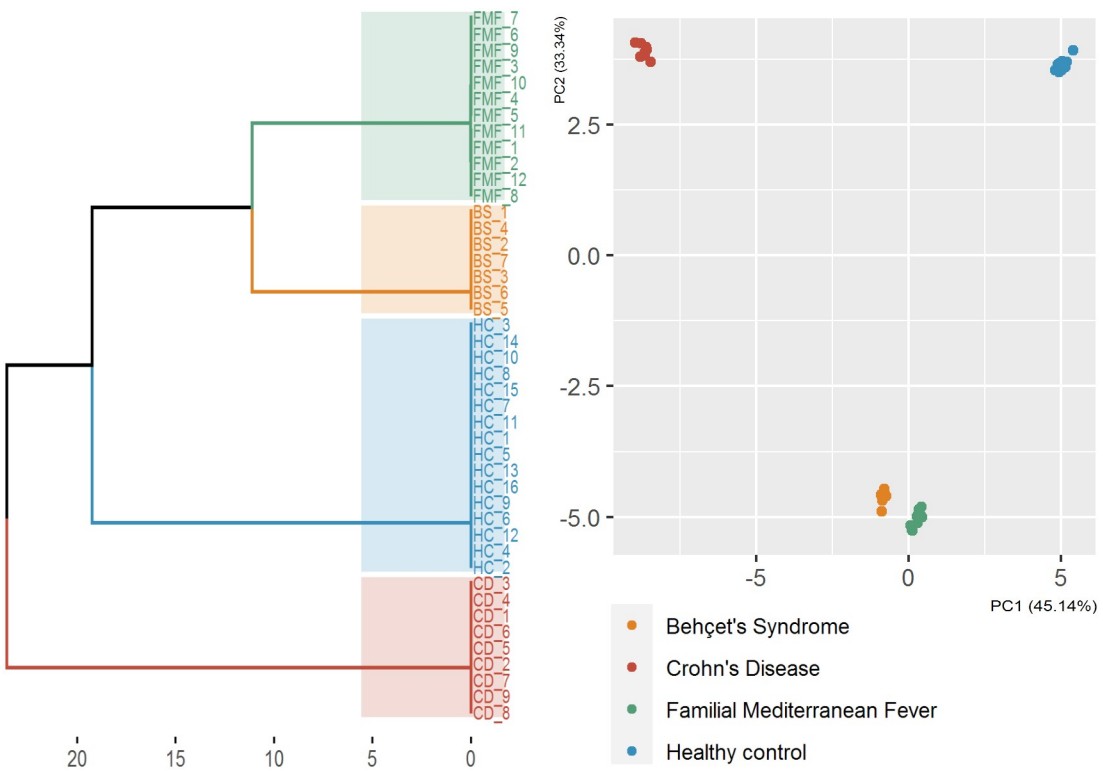

**Fig 1. The PCoA model was established, corresponding to the four groups (Healthy Controls, "HC" blue; Behçet's Syndrome, "BS" orange; Crohn's Disease, "CD" red and Familial Mediterranean Fever, "FMF" green) and hierarchical cluster analysis by Bray-Curtis dissimilarity distance.**

*Prevotellaceae* and *Lachnospiraceae*. FMF patient's microbiota consisted of *Prevotellaceae* in dominancy, followed by *Ruminococcaceae*, *Lachnospiraceae* and *Enterobacteriaceae*, *Veionellaceae*.

We considered that predominant families were *Veillonellaceae*, *Succinivibrionaceae* in BS group while *Bacteroidaceae* was nearly absent in BS cases. Contrary to which, *Bacteroidaceae* was dominant in the healthy group while *Veillonellaceae*, *Succinivibrionaceae* were in limited numbers. Fig 5 shows the heatmap of samples while Fig 6 shows predominant families appeared to be significant for BS.

At the genus level most abundant genera in BS group were; *Succinivibrio*, *Prevotella*, *Mitsuokella*, *Anaerostipes*, *Catonella*, *Fusicatenibacter*, *Lachnoanaerobaculum*, *Marvinbryantia*, *Moryella*, *Robinsoniella*, *Tyzzerella* and *Faecalibacterium*. The top five genera in HC group were; *Bacteroides*, *Prevotella*, *Faecalibacterium*, *Parabacteroides*, *Phascolarctobacterium*. Top five genera in CD group were; *Dialister*, *Bacteroides*, *Prevotella*, *Barnesiella* and *Faecalibacterium*. The top five genera in FMF group were; *Prevotella*, *Faecalibacterium*, *Bacteroides*, *Roseburia*, *Catenibacterium*. *Prevotella* and *Faecalibacterium* genera considered as common genera since they were found in all study groups presenting different colonisation levels. Succinivibrio and Mitsuokella were evaluated as "BS specific genera" because, they were isolated predominantly in BS group while they were not isolated from other study groups.

The top 5 species in all groups were as follows; *Prevotella copri*, *Faecalibacterium prausnitzii*, *Succinivibrio dextrinosolvens*, *Mitsuokella jalaludinii*, *Dialister invisus* in BS group, *Bacteroides eggerthii*, *Bacteroides vulgatus*, *Prevotella copri*, *Faecalibacterium prausnitzii*, *Phascolarctobacterium faecium* in HC group, *Bacteroides vulgatus*, *Prevotella copri*, *Prevotella*

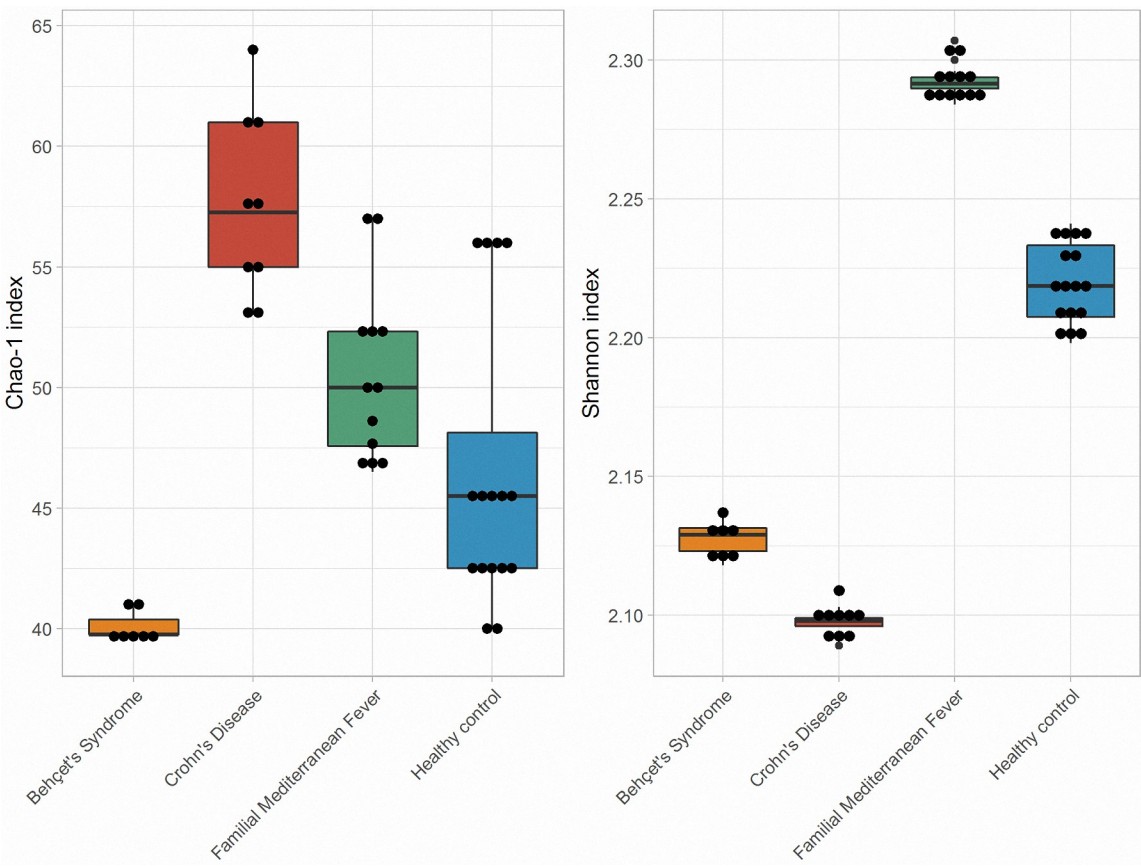

**Fig 2. Chao-1 richness and Shannon diversity indexes comparisons of Behçet's syndrome and other groups.**

*stercorea, Faecalibacterium prausnitzii, Gemmiger formicilis* in FMF group, *Bacteroides thetaiotaomicron, Barnesiella intestinihominis, Faecalibacterium prausnitzii, Prevotella stercorea, Dialister invisus* in CD group. We could not consider a "BS specific" or predominant species since relative abundance were similar in all groups on species level. The dominant composition of gut microbiota of BS patients was listed in Table 2.

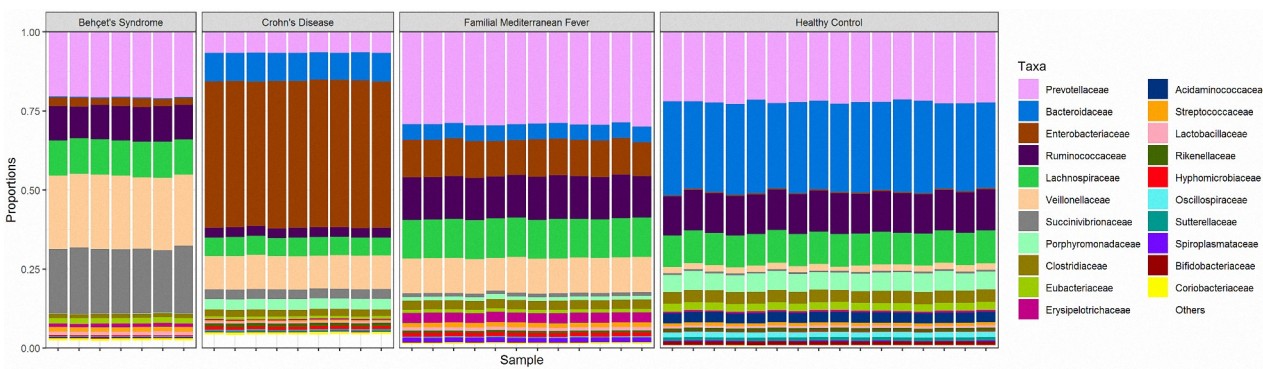

**Fig 3. The most abundant taxa at the family level,** *Veillonellaceae* **(light orange) and** *Succinivibrionaceae* **(light gray) were more abundant in the fecal microbiota of the Behçet's Syndrome group compared with that of the healthy control group, while** *Bacteroidaceae* **was consistently less abundant (p<0.001).** Compared with the healthy group, the FMF and Crohn's group had a significantly higher abundance of *Enterobacteriaceae* (p<0.001). Crohn's group significantly reduced abundance of *Ruminococcaceae* compared with the healthy group (p<0.001).

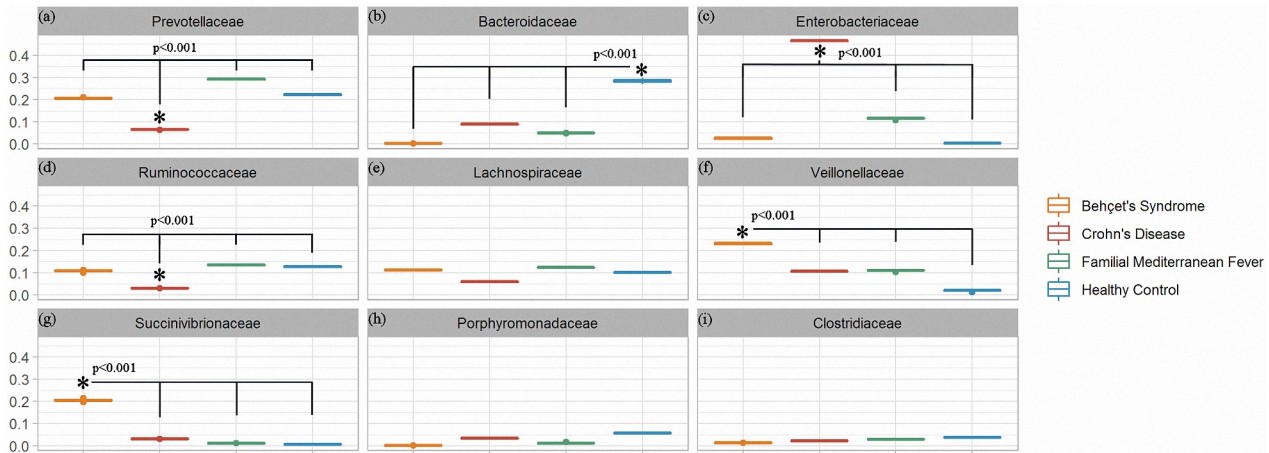

**Fig 4.** The comparison (* = p<0.001) of nine of the most abundant [(**a**) higher to (**i**) lower] taxa for presenting a relationship with healthy controls. Only (**g**) Succinivibrionaceae abundance in Behçet's Sydrome patients' gut microbiota is the higher than other groups. Also, (**f**) Veillonellaceae family in Behçet's Syndrome patients' gut microbiota is higher than other groups, it is lower compared to (**g**) Succinivibrionaceae. The FMF and Crohn's patients had a significantly higher abundance of (**c**) Enterobacteriaceae than other groups. Crohn's group significantly reduced abundances of (**a**) Prevotellaceae and (**d**) Ruminococcaceae families compared with the healthy group. (**b**) Bacterioidaceae family is higher in the healthy group than others. (**e**) Lachnospiraceae, (**h**) Porphyromonadaceae and (**i**) Clostridiaceae families have no statistically different among groups.

## Discussion

In our study, we examined the fecal microbial flora of 7 BS patients with uveitis. We identified dominant microbial communities on phylum, class, order, family, genus and species levels. In BS patients, the fecal microbiota consisted mainly of Firmicutes as a phylum, Clostridia as a class, Clostridiales as an order and *Prevotella copri* as a species.

*Veionellaceae*, *Succinivibrionaceae* families and *Succinivibrio*, *Mitsuokella* genera were found to be dominant in the gut microbiota of BS patients who were fed by different diet and had different genetical factors. Technically, relative abundances were similar in each 7 BS case, the range is narrow and the variance is low. Relative abundances of members of study groups

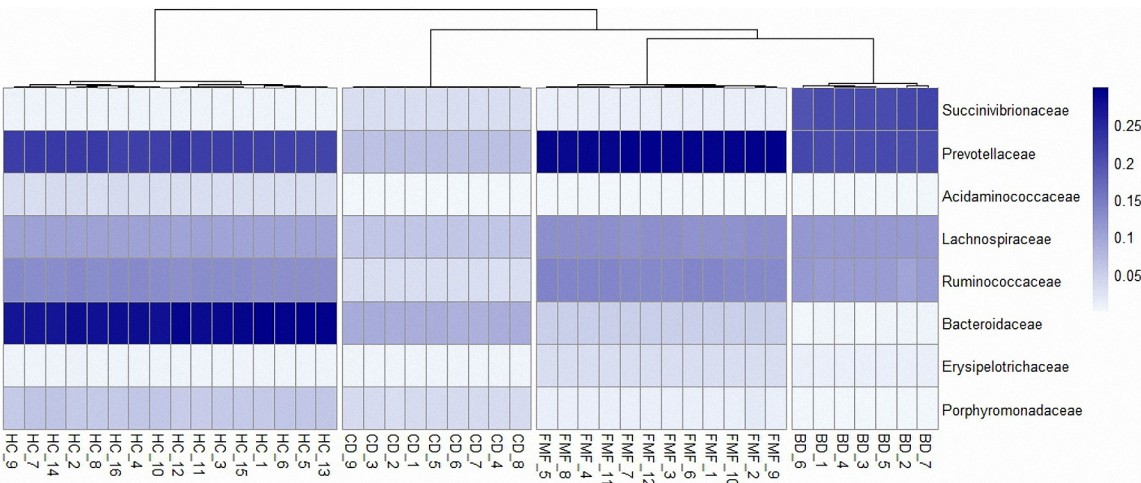

**Fig 5. Heatmap of samples (the most abundant taxa with their relative abundance) based on information at the genus level.** Columns present the grouped cases (with hierarchical cluster tree according to Bray-Curtis distance), and rows represent the taxa. The values (color key) in the heatmap represent the relative abundance.

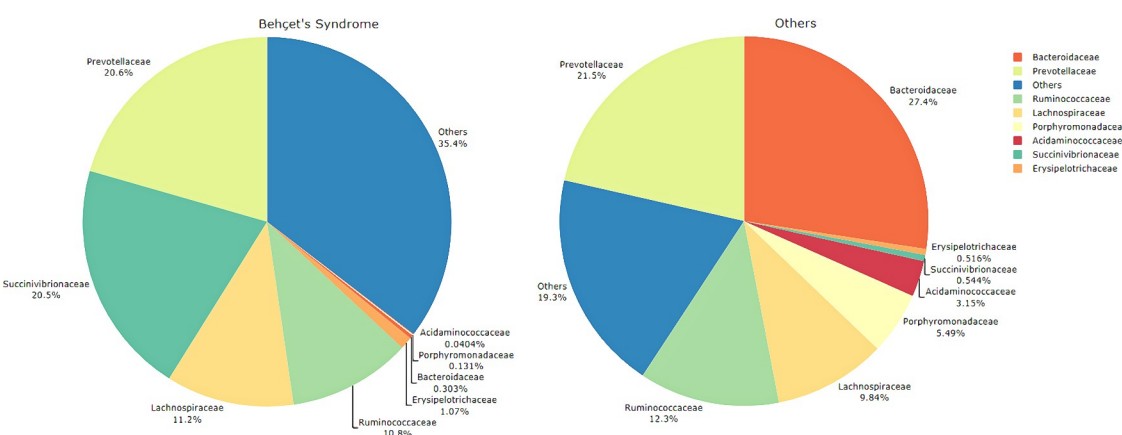

**Fig 6. Predominant families appeared to be significant for BS.**

were technically homogeneous. Therefore, we think *Veionellaceae*, *Succinivibrionaceae* families and *Succinivibrio*, *Mitsuokella* genera as dominant taxa in the gut microbiota of BS cases. The role of microbiota in the pathogenesis of BS is not clearly defined. There are limited number of studies focused on oral and salivary microbial community in BS patients [27–29]. Molecular mimicry based on sequence homology between microbial and human heat-shock protein (HSP) peptides trigger autoimmune responses in patients with BS patients. Several autoantigens have been observed, including the HSP60 kDa and HSP70 kDa proteins, S antigen, interphotoreceptor retinoid-binding protein (IRBP), α-tropomyosin, and αβ-crystallin [30]. Therefore, a greater understanding the role of the microbial community in the pathogenesis of BS will probably lead to improved development of therapeutic strategies [31].

Consolandi et al, [32] presented 22 BS patients who exhibited a specific microbiome signature in their fecal microbiota. However in that study, just one hypervariable region (V4) was sequenced using Roche 454 technology. They found that BS patients were characterized by a low biodiversity and by a depletion of key butyrate-producing bacteria in their gut microbiota. Butyrate was evaluated as an impaired microbial metabolite in that group of BS patients. Butyrate impairment could be the reason of reduced T-regulator cell mediated control, and promotes powerful immunopathological T cell responses [32]. Healthy gut microbiota is dominated by microorganisms belonging to Bacteroidetes phylum [11]. Bacteroidetes phylum was dominant in our HC group, too. This group of bacteria produces short-chain fatty acids, such as butyrate, which has anti-inflammatory effects.

In our study, we found Firmicutes and Proteobacteria were dominant phyla in BS patients presenting inflammatory effects in the gastrointestinal system. Shin et al. reviewed that an increased prevalence of the bacterial phylum Proteobacteria is a marker for an unstable

**Table 2. Dominant microbial communities in BS patients with uveitis.**

| | Dominant microbial communities in BS patients with uveitis |
|---|---|
| **Pyhlum** | Firmicutes, Proteobacteria |
| **Class** | Clostridia, Gammaproteobacteria, Gammaproteobacteria, Negativicutes |
| **Order** | Clostridiales, Bacteroidales, Selenomonadales, Aeromonadales |
| **Family** | *Veillonellaceae*, *Succinivibrionaceae*, *Prevotellaceae*, *Lachnospiraeae* |
| **Genus** | *Succinivibrio*, *Mitsuokella*, *Prevotella* |
| **Species** | *Prevotella copri* |

microbial community (dysbiosis) and a potential diagnostic criterion for inflammatory disorders [33]. Sheridan et al. noted that Firmicutes possess large genomes with extremely high numbers of predicted carbohydrate-active enzymes (CAZymes). These CAZyme genes are located in the genome adjacent to genes encoding regulators and carbohydrate transport functions, forming multiple polysaccharide utilization loci (PULs) predicting inflammation [34].

Ye et al, recently reported that Bacteroidales, Firmicutes and Proteobacteria were dominant phyla in BD. Their results showed that sulfate-reducing bacteria (SRB) were significantly enriched in BD, whereas methanogens and butyrate-producing bacteria were negatively associated with the BD patients [29].

As for diversity and dysbiosis in FMF and CD; In patients with FMF, similar to BS, Firmicutes was the dominant phyla, while Proteobacteria was dominant in CD group. *Bacteroidetes* spp. was the dominant genera for both CD and FMF patients, which was followed by *Alistipes* spp. *Alistipes* species have been shown to cause colitis and site specific tumors in animal studies [35]. We also found *Enterobacteriaceae* colonization predominantly in CD patients. This family is considered as a pathogenic species, rather than a commensal one with harmful effects on human metabolism.

Shimizu J et al, have found that there was a significant difference in 11 bacterial taxa between BS patients and normal individuals. At genus level they have found *Bifidobacterium* and *Eggerthella* increased significantly and *Provetella* decreased in BS patients compared with normal individuals in Japanese population [36]. However, at genus level we found *Succinivibrio* and *Mitsuokella* dominancy in gut microbiota in our Turkish BS patients.

The dominant species was *Prevotella copri* in our BS study group. The association of *Prevotella* with BS is still controversial. The pustular lesions of BS patients were found to be infected by *Prevotella* spp. [19]. In another study, *Prevotella intermedia* was isolated from subgingival plaque samples of BS patients [37]. On the contrary, *Prevotella* was significantly lower in the oral microbiota of patients with BS than in controls [38]. *Prevotella copri* is an inflammatory bacteria which was found to be in concordance with arthritis [39]. Inflammation of the gut mucosa, mediated by *Prevotella* spp, promotes systemic dissemination of inflammatory mediators, increased intestinal permeability and translocation of bacterial products, which amplify and promote systemic inflammation [40]. MaedaY et al, found that fecal microbiota of patients with early rheumatoid arthritis (RA) was dominated by *Prevotella copri* leading to dysbiosis and autoreactive T cell activation [41, 42]. Pianta et al, have identified an HLA-DR presented peptide from *Prevotella copri* which stimulates Th1 responses in 42% of new-onset RA patients [43]. Wen et al, have reported that ankylosing spondylitis patients demonstrated increased levels of *Prevotella melaninogenica*, *Prevotella copri* and other *Prevotella* spp. in their gut microbiota [44]. In a recent study of Shimizu J et al, has shown a relative abundance of *Eggerthella lenta*, *Acidaminococcus* species, *Lactobacillus mucosae*, *Bifidobacterium bifidum*, *Lactobacillus iners*, *Streptococcus* species, and *Lactobacillus salivarius* in patients with BS [45].

During the course of the disease, BS can involve the mucosa, skin, or eyes, vessels, nervous and gastrointestinal system [22, 23, 46]. Several cluster and association studies reported clusters of common overlapping involvements. In a retrospective observational study on 295 BS patients, Bitik et al reported that BS patients with posterior uveitis had a significantly higher risk of having neurological involvement compared to those without posterior uveitis [47]. Similarly, Yan et all reported that ocular involvement was more prevalent in parenchymal neuro-BS compared with age and sex matched BS [48]. Although ocular involvement is a "warning sign" for predisposition to neurological involvement in BS, the pathogenetic mechanisms of the concomitant occurrence of these manifestations have not been described. On the other hand, Suwa et al found a negative association of eye involvement with genital ulceration and gastrointestinal involvement at the early stage of disease [49]. Similarly, Hussein et al reported

that genital ulcers and systemic vasculitis were protective factors for the development of vision threatening eye disease [50]. We focused on BS patients with eye involvement since, each involvement has its own demographic and clinical characteristics, and also medical treatment and prognosis of each manifestation differs greatly [46]. Oezguen N et al. have subjected neuro-Behçet's disease for the analysis of gut microbiota. They concluded that microbiota stratification identifies disease specific alterations. They identified *Prevotella* and *Bacteroides* dominated subsample clusters in neuro-Behçet's disease cohorts [51].

One of the limitations of our study is the lack of mycobiome and virome analysis of the fecal samples. Mycobial and viral agents have significant effects on the innate immune system as well. In this study we could only investigate the bacterial diversity. Another limitation is that 5 patients in BS group were on immunosuppressive treatment. However, despite being on immunosuppressive therapy these patients had still active uveitis attacks at the time of fecal sample collection. Moreover, their fecal microbiome showed technically homogeneous OTUs values; for this reason, we do not think that immunosuppressive therapy had an important impact on our results. It would be crucial that future studies investigate the polymicrobial interactions in the BS setting. Focusing on the bacterial community alone, limits the possibility of discerning possible associations between the fecal microbiota. The majority of the studies have focused primarily on the bacterial community of the microbiome such as we performed in the present study. However, the fungal portion of the microbiome (mycobiome) should be included into analyses. Concerted efforts should be directed to the systems biology approach that links the microbiota, their metabolites and BS pathogenesis. There is a need to conduct studies in the BS patients that use a comprehensive systems biology approach to identify the link between the mycobiome, bacteriome, virome and their metabolites. There is limited data about the fungal community of the gut microbiome in BS [29]. Ye at al. found that *Atkinsonella texensis*, *Trichoderma parareesei*, *Colletotrichum orbiculare*, *Exophiala mesophila*, *Candida parapsilosis*, *Claviceps paspali*, *Drechslerella stenobrocha*, *Shiraia* spp. are dominant fungal species in the fecal microbiome of BS patients.

With the recent advances in next-generation sequencing analysis and identification of the multitude of microorganisms comprising the human microbiome has become one of the most intensely studied areas in both health and disease. The altered microbial community in the gut microbiota of BS patients warrants further research in this area. Dysbiosis leads to the release of microbial peptides, other molecules that amplify a state of chronic inflammation. Environmental factors also contribute to triggering inflammation such as smoking, diet, infectious pathogens and medications, stress, lack of sleep and exercise [52].

Questions about the primitive district in which dysbiosis will occur remain. The real impact of dysbiosis on the course of BS and to conceive therapeutic strategies to counteract microbiome- driven inflammation should be clearly defined.

## Acknowledgments

The authors would like to thank to Msc. Bio. Yoruk Divanoglu for his technical assistance into next generation sequencing.

## Author Contributions

**Data curation:** Duygu Tecer, Feride Gogus, Murat Hasanreisoglu, Çagri Ergin.

**Formal analysis:** Ayse Kalkanci, Çagri Ergin, Ramazan Kozan, Kadir Serdar Diker.

**Funding acquisition:** Feride Gogus.

**Investigation:** Feride Gogus, Ayse Kalkanci, Murat Hasanreisoglu, Tarkan Karakan, Seda Coban.

**Methodology:** Duygu Tecer, Feride Gogus, Ayse Kalkanci, Merve Erdogan, Çagri Ergin, Tarkan Karakan, Seda Coban, Kadir Serdar Diker.

**Software:** Çagri Ergin, Ramazan Kozan.

**Validation:** Ramazan Kozan.

**Writing – original draft:** Ayse Kalkanci, Kadir Serdar Diker.

**Writing – review & editing:** Ayse Kalkanci, Ramazan Kozan, Kadir Serdar Diker.

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
