## [Decision Letter · Decision Letter 0]

13 May 2020

PONE-D-19-35822

Succinivibrio and Mitsuokella are Dominant Generas in Fecal Microbiota of Behçet’s Syndrome Patients with Mucocutanous and Uveitis Involvement

PLOS ONE

Dear Dr Kalkanci,

Thank you for submitting your manuscript to PLOS ONE. After careful consideration, we feel that it has merit but does not fully meet PLOS ONE’s publication criteria as it currently stands. Therefore, we invite you to submit a revised version of the manuscript that addresses the points raised during the review process.

I would like to sincerely apologise for the delay you have incurred with your submission. We have now received two completed reviews; their comments are available below.

Both reviewers have raised several concerns about the selection of patients and the methodology of this study that need to be addressed in a revision. Please revise the manuscript to address all the reviewer's comments in a point-by-point response in order to ensure it is meeting the journal's publication criteria. Please note that the revised manuscript will need to undergo further review, we thus cannot at this point anticipate the outcome of the evaluation process.

We would appreciate receiving your revised manuscript by Jun 26 2020 11:59PM. To enhance the reproducibility of your results, we recommend that if applicable you deposit your laboratory protocols in protocols.io, where a protocol can be assigned its own identifier (DOI) such that it can be cited independently in the future. For instructions see: http://journals.plos.org/plosone/s/submission-guidelines#loc-laboratory-protocols

We look forward to receiving your revised manuscript.

Kind regards,

Miquel Vall-llosera Camps

Associate Editor

PLOS ONE

2. In your Methods section, please state where the participants were recruited for your study.

4. We note that you are reporting an analysis of a microarray, next-generation sequencing, or deep sequencing data set. PLOS requires that authors comply with field-specific standards for preparation, recording, and deposition of data in repositories appropriate to their field. Please upload these data to a stable, public repository (such as ArrayExpress, Gene Expression Omnibus (GEO), DNA Data Bank of Japan (DDBJ), NCBI GenBank, NCBI Sequence Read Archive, or EMBL Nucleotide Sequence Database (ENA)). In your revised cover letter, please provide the relevant accession numbers that may be used to access these data. For a full list of recommended repositories, see http://journals.plos.org/plosone/s/data-availability#loc-omics or http://journals.plos.org/plosone/s/data-availability#loc-sequencing.

5. Please include your tables as part of your main manuscript and remove the individual files. Please note that supplementary tables (should remain/ be uploaded) as separate "supporting information" files.

7. Please upload a copy of Figure 6, to which you refer in your text on page 7. If the figure is no longer to be included as part of the submission please remove all reference to it within the text.

8. Please upload a new copy of Figure 2 as the detail is not clear. Please follow the link for more information: https://blogs.plos.org/plos/2019/06/looking-good-tips-for-creating-your-plos-figures-graphics/

**Comments to the Author**

1. Is the manuscript technically sound, and do the data support the conclusions?

Reviewer #1: Partly

Reviewer #2: Yes

2. Has the statistical analysis been performed appropriately and rigorously? 

Reviewer #1: No

Reviewer #2: Yes

3. Have the authors made all data underlying the findings in their manuscript fully available?

Reviewer #1: No

Reviewer #2: Yes

4. Is the manuscript presented in an intelligible fashion and written in standard English?

Reviewer #1: No

Reviewer #2: Yes

5. Review Comments to the Author

Reviewer #1: The authors conducted a metagenomic analysis in patients with BD and mined the species Prevotella copri from the data as a BD characteristic microbe. BD is rare and often difficult to treat even now and the finding may become very important data for the patients and treating physicians in the near future.

Major:

1. In the introduction section, the authors should describe detailed information of the differences in the “immunological pathologies” between BD patients with and without uveitis. We did not understand why the authors selected BD patients with uveitis for this study. If the authors need to use the data of FMF and CD as controls in this study, commonalities and differences in the “immunological pathologies” between BD and FMF, and between BD and CD would be helpful for us to understand more accurately. Discussion of the relationships between previous data of the immunological pathologies in patients with BD and metagenomic data of this study would be also important for the precise understanding.

2. The authors described their metagenomic data using OTU numbers in their figures and tables. Nonetheless, in their statistical analyses, the authors compared the metagenomic data using the taxonomic abundance. In the materials and methods section, the authors described that they used Kruskal-Wallis rank sum test to find significant differences in taxonomic abundance between patients and normal individuals (page 5). In the discussion section, the authors described that OTUs of this study groups were technically homogeneous (page 9). These descriptions were very confusing for us. If we understand it right, the authors used taxonomic abundance data for all statistical analyses of this study. We suggest that the authors utilize relative abundance titers of the metagenomic data for the comparison in the figures and tables with the statistical data. At least the figures and tables need the data of statistical analyses.

3. The authors should provide us with BD patient demographical and clinical data with the statistical analyses. Gender (Front Immunol 2017; 8: 754.), aging (Cell Mol Life Sci 2018; 75: 129-148.), and medications (nonantibiotics, Nature 2018; 555: 623-8.) were reported to associate with gut microbe alteration. If the authors need to use the data of FMF and CD as controls in this study, the authors should provide us with the patients' data.

4. High titers of alpha diversity in gut microbe composition were considered as a healthy state (Hepatology 2017; 65: 451-64.). The authors should describe the alpha diversity index titers of the 4 subgroups in this study with the detailed statistical analyses.

5. We need several references in the next two sentences for the precise understanding.

(1) Page 9: The molecular mimicry based on sequence homology between microbial and human peptides triggers autoimmune responses in BS patients.

(2) Page 10: We found Firmicutes and Proteobacteria were dominant phyla in BS patients presenting inflammatory effects in the gastrointestinal system.

6. The authors should send the manuscript to the natives for revising the English.

Minor:

1. We were wondering if the authors could provide us with the information of their minor revisions of the elusion steps in DNA extractions (Page 4).

2. The authors should spell out the term of Treg in Page 9. Several references in the sentence would be helpful for us.

Reviewer #2: The aim of this study was to investigate the gut microbiota of Behçet’s Syndrome patients with uveitis involvement and compare the gut microbiota of BS with that of Familial Mediterranean Fever and Crohn’s Disease. However, the following issues should be addressed.

1. The disease profiles of BS patients included in this study are unclear. Authors need to prepare the “Table” of patient’s data, such as the patients with BS (oral ulcer, skin lesion, genital ulcer, and so on).

2. Authors should also mention whether the patients were with active or inactive BS. When was the fecal sample obtained relative to the onset of an attack?

3. Did the patients included in this study receive medication before the fecal sample obtained? Immunomodulatory drugs and steroids would impact on the gut microbiota composition.

4. The detailed information of control was also unclear. The human gut microbiota composition will differ from the age, sex, life style, diet and inflammatory status of donors (see PMID: 24705962 and 20498852). Thus, author should provide more profiles of patients and healthy controls, such as their age and sex. And author should mention whether the healthy control were with diabetes, cardiovascular diseases, systemic disease and other inflammatory diseases.

5. The data regarding the abundant taxa at the family level in four groups suggested Succinivibrio and Mitsuokella were dominant and signature families, whereas Bacteroides was absent in BS patients. However, the statistical analysis were without correction. The statistical analysis in the genera and species levels also required correction.

6. The authors are aware of the limitations of this study. However, mono-colonization of dominant species or fecal transplantation from BS patients in germ-free mice would also be informative.

7. The OTU clustering methods need to be clarified. The version of the software and algorithms used also need to be indicated. It is also unclear how the authors performed taxonomic assignment or what database they used.

8. Minor issue, see Page 3, lines 17, "Behçet's syndrome" should be used as abbreviation "BS"

9. Minor issue, see Page 11, lines 1, "Prevotella" should be used in italic.

6. PLOS authors have the option to publish the peer review history of their article (what does this mean?). If published, this will include your full peer review and any attached files.

Reviewer #1: No

Reviewer #2: No

---

## [Author Response · Author response to Decision Letter 0]

18 Jun 2020

PONE-D-19-35822

Succinivibrio and Mitsuokella are Dominant Generas in Fecal Microbiota of Behçet’s Syndrome Patients with Uveitis

PLOS ONE

Dear editor,

We revised the manuscript to address all the reviewer's comments in a point-by-point response. This rebuttal letter that responds to each point raised by the academic editor and reviewers. 

Ayse Kalkanci

Corresponding author

Editor’s comments:

Style requirements were completed. DOI numbers were added if exists.

2. In your Methods section, please state where the participants were recruited for your study.

It was stated in methods section “Participants were recruited to our study at Gazi University Medical Faculty, Physical Medicine and Rehabilitation Department, Division of Rheumatology, Ankara, Turkey”. 

Informed written consent was obtained from each participant included in this study, according to the declaration of Helsinki.

4. We note that you are reporting an analysis of a microarray, next-generation sequencing, or deep sequencing data set. PLOS requires that authors comply with field-specific standards for preparation, recording, and deposition of data in repositories appropriate to their field. Please upload these data to a stable, public repository (such as ArrayExpress, Gene Expression Omnibus (GEO), DNA Data Bank of Japan (DDBJ), NCBI GenBank, NCBI Sequence Read Archive, or EMBL Nucleotide Sequence Database (ENA)). In your revised cover letter, please provide the relevant accession numbers that may be used to access these data. For a full list of recommended repositories, see http://journals.plos.org/plosone/s/data-availability#loc-omics or http://journals.plos.org/plosone/s/data-availability#loc-sequencing.

Nucleotide suquences of 1797 taxa were deposited in NCBI GenBank with accession numbers MT573971 to MT575767 (file SUB7562864). 

 5. Please include your tables as part of your main manuscript and remove the individual files. Please note that supplementary tables (should remain/ be uploaded) as separate "supporting information" files.

Tables were included as a part of our manuscript. 

ORCID iD for the corresponding author 0000-0003-0961-7325

7. Please upload a copy of Figure 6, to which you refer in your text on page 7. If the figure is no longer to be included as part of the submission please remove all reference to it within the text.

Reference of “Figure 6” was removed on page 7 and within the text. 

8. Please upload a new copy of Figure 2 as the detail is not clear. Please follow the link for more information: https://blogs.plos.org/plos/2019/06/looking-good-tips-for-creating-your-plos-figures-graphics/

Figure 2 was revised and uploaded. 

Reviewer 1’s comments;

Reviewer #1: The authors conducted a metagenomic analysis in patients with BD and mined the species Prevotella copri from the data as a BD characteristic microbe. BD is rare and often difficult to treat even now and the finding may become very important data for the patients and treating physicians in the near future.

Major:

1. In the introduction section, the authors should describe detailed information of the differences in the “immunological pathologies” between BD patients with and without uveitis. We did not understand why the authors selected BD patients with uveitis for this study. If the authors need to use the data of FMF and CD as controls in this study, commonalities and differences in the “immunological pathologies” between BD and FMF, and between BD and CD would be helpful for us to understand more accurately. Discussion of the relationships between previous data of the immunological pathologies in patients with BD and metagenomic data of this study would be also important for the precise understanding.

Introduction and discussion were both re-written including new comparative data BD&CD and BD&FMF. New references were added. 

2. The authors described their metagenomic data using OTU numbers in their figures and tables. Nonetheless, in their statistical analyses, the authors compared the metagenomic data using the taxonomic abundance. In the materials and methods section, the authors described that they used Kruskal-Wallis rank sum test to find significant differences in taxonomic abundance between patients and normal individuals (page 5). In the discussion section, the authors described that OTUs of this study groups were technically homogeneous (page 9). These descriptions were very confusing for us. If we understand it right, the authors used taxonomic abundance data for all statistical analyses of this study. We suggest that the authors utilize relative abundance titers of the metagenomic data for the comparison in the figures and tables with the statistical data. At least the figures and tables need the data of statistical analyses.

We used relative abundance titers of the metagenomic data for the comparison in the figures and tables with the statistical data. Statistical analyses were added into the figures and tables. Materials and methods section was re-written for statistical analysis. 

3. The authors should provide us with BD patient demographical and clinical data with the statistical analyses. Gender (Front Immunol 2017; 8: 754.), aging (Cell Mol Life Sci 2018; 75: 129-148.), and medications (nonantibiotics, Nature 2018; 555: 623-8.) were reported to associate with gut microbe alteration. If the authors need to use the data of FMF and CD as controls in this study, the authors should provide us with the patients' data.

Demographical and clinical characteristics of participants at the time of fecal sample collection were summaized at table 1

4. High titers of alpha diversity in gut microbe composition were considered as a healthy state (Hepatology 2017; 65: 451-64.). The authors should describe the alpha diversity index titers of the 4 subgroups in this study with the detailed statistical analyses.

Described. 

5. We need several references in the next two sentences for the precise understanding.

(1) Page 9: The molecular mimicry based on sequence homology between microbial and human peptides triggers autoimmune responses in BS patients.

A new reference numbered by 16 was added “Hedayatfar A. Behçet's Disease: Autoimmune or Autoinflammatory?. J Ophthalmic Vis Res 2013; 8 (3): 291‐293”. 

(2) Page 10: We found Firmicutes and Proteobacteria were dominant phyla in BS patients presenting inflammatory effects in the gastrointestinal system.

Two new references numbered by 18 and 19 were added “18. Shin NR, Whon TW, Bae JW. Proteobacteria: microbial signature of dysbiosis in gut microbiota. Trends Biotechnol 2015; 33 (9): 496-503” and “19. O Sheridan P, Martin JC, Lawley TD, et al. Polysaccharide utilization loci and nutritional specialization in a dominant group of butyrate-producing human colonic Firmicutes. Microb Genom 2016; 2 (2): e000043”. 

6. The authors should send the manuscript to the natives for revising the English.

Revised.

Minor:

1. We were wondering if the authors could provide us with the information of their minor revisions of the elusion steps in DNA extractions (Page 4).

Minor revision at the elution step was performed by using 30 µl elution buffer.

2. The authors should spell out the term of Treg in Page 9. 

“T regulator cell= Treg" 

Reviewer #2: The aim of this study was to investigate the gut microbiota of Behçet’s Syndrome patients with uveitis involvement and compare the gut microbiota of BS with that of Familial Mediterranean Fever and Crohn’s Disease. However, the following issues should be addressed.

1. The disease profiles of BS patients included in this study are unclear. Authors need to prepare the “Table” of patient’s data, such as the patients with BS (oral ulcer, skin lesion, genital ulcer, and so on).

Demographical and clinical characteristics of participants at the time of fecal sample collection were summarized at table 1. Age were similar between groups (p: 0.098). 

Table 1. Demographical and clinical characteristics of participants at the time of fecal sample collection

 Behçet’s Syndrome 

(n:7) Crohn’s disease 

(n:9) Familial Mediterranean Fever (n:12) Healthy Controls

(n:16)

Age, years 35.57 ± 6.60 35.00 ± 5.27 32.17 ± 8.64 39.38 ± 7.69

Men/women 5/2 3/6 6/6 6/10

Disease characteristics 

Oral aphthosis, n (%) 5 (71.43%) 1 (11.11%) 0 0

Genital ulcers, n (%) 1 (14.29%) 0 0 0

Skin lessions, n (%) 3 (42.86%) 1 (11.11%) 0 0

Uveitis, n (%) 7 (100%) 0 0 0

Gastrointestinal system involvement, n (%) 0 9 (100%) 0 0

Central nervous system involvement, n (%) 1 (14.29%) 0 0 0

Vascular involvement, n (%) 0 0 0 0

Arthritis, % 0 2 (22.22%) 2 (16.67) 0

Pathergy positivity 2 (28.57%) NA NA NA

HLA-51 4 (57.14%) NA NA NA

Medications 

Colchicine, n (%) 6 0 12 (100%) 0

Steroid, n (%) 0 0 0 0

5-ASA/sulfasalazine, n (%) 0 0 0 0

Cyclosporine, n (%) 1 0 0 0

Azathioprine, n (%) 4 9 0 0

Methotrexate, n (%) 0 0 0 0

Infliximab, n (%) 0 0 0 0

 NA, not applicable

2. Authors should also mention whether the patients were with active or inactive BS. When was the fecal sample obtained relative to the onset of an attack?

The fecal samples of patients with BS were obtained at the active uveitis attack. 

3. Did the patients included in this study receive medication before the fecal sample obtained? Immunomodulatory drugs and steroids would impact on the gut microbiota composition.

Informations about medication at the time of sample collection were summarized at table 1. 

4. The detailed information of control was also unclear. The human gut microbiota composition will differ from the age, sex, life style, diet and inflammatory status of donors (see PMID: 24705962 and 20498852). Thus, author should provide more profiles of patients and healthy controls, such as their age and sex. And author should mention whether the healthy control were with diabetes, cardiovascular diseases, systemic disease and other inflammatory diseases.

16 healthy controls; 6 male (39-50y) and 10 female (24-52y) matched for age and sex. Patients were excluded from the study if they had any of the following: patient suspected to have BD clinically but did not fulfil the inclusion criteria; pregnancy, chronic diseases, psychiatric disorders, cancer, diabetes, cardiovascular diseases, systemic disease and other inflammatory diseases, dependence on alcohol or other substances. Dietary behaviors were similar in BS, FMF, CD and HC groups. 

Demographical characteristics including age and sex of all participants were summarized at table 1. Individuals without systemic disease and inflammatory disease were included as healthy controls.

5. The data regarding the abundant taxa at the family level in four groups suggested Succinivibrio and Mitsuokella were dominant and signature families, whereas Bacteroides was absent in BS patients. However, the statistical analysis were without correction. The statistical analysis in the genera and species levels also required correction.

All data was corrected and marked.

6. The authors are aware of the limitations of this study. However, mono-colonization of dominant species or fecal transplantation from BS patients in germ-free mice would also be informative.

Information about the dominant fungal species obtained from BS patients was added into discussion. (Ye Z, Zhang N, Wu C, Zhang X, Wang Q, Huang X, Du L, Cao Q, Tang J, Zhou C, Hou S, He Y, Xu Q, Xiong X, Kijlstra A, Qin N, Yang P. A metagenomic study of the gut microbiome in Behcet's disease. Microbiome 2018; 6 (1): 135. 

7. The OTU clustering methods need to be clarified. The version of the software and algorithms used also need to be indicated. It is also unclear how the authors performed taxonomic assignment or what database they used.

Obtained sequences were transferred into Operational Taxonomic Units (OTUs). Similarity of 99% was accepted as same phylotype. Each value was expressed as mean ± SD and a p value less than 0.05 was considered significant. 

Statistical analysis was performed using R (Ver 3.6.0,, GPL-3 license). The relative abundances (expressed as parts per unit, calculated with ade4 packet in R, were directly compared in each taxonomic (for different phylum, genus, species) levels. vegan package was used for alpha biodiversity. Alpha diversity is defined as the diversity within a community and is mainly measured by Chao 1 diversity indexes. Chao 1 index estimates richness in numbers of taxa frequencies. The microbial diversities data were tested by using Wilcoxon rank sum test in Past 3.20 (Øyvind Hammer, 2018). Based on relative abundance features, R packages FactoMineR and factoextra were used for dendrogram (Bray-Curtis similarity) and taxa distribution visualizations with groups while ggfortify and ggplot2 were used for PCA.

8. Minor issue, see Page 3, lines 17, "Behçet's syndrome" should be used as abbreviation "BS"

Corrected

9. Minor issue, see Page 11, lines 1, "Prevotella" should be used in italic.

 Corrected

---

## [Editor Report · Decision Letter 1]

15 Jul 2020

PONE-D-19-35822R1

Succinivibrio and Mitsuokella are Dominant Generas in Fecal Microbiota of Behçet’s Syndrome Patients with Uveitis

PLOS ONE

Dear Dr. Ayse Kalkanci,

Thank you for submitting your manuscript to PLOS ONE. After careful consideration, we feel that it has merit but does not fully meet PLOS ONE’s publication criteria as it currently stands. Therefore, we invite you to submit a revised version of the manuscript that addresses the points raised during the review process.

Please see below for the reviwer`s comments.

We look forward to receiving your revised manuscript.

Kind regards,

Noboru Suzuki

Academic Editor

PLOS ONE

Journal Requirements:

Additional Editor Comments (if provided):

Major:

1. The authors described that they found significant differences in the bacterial relative abundance of family level with the p values (Fig 3) between diseases and healthy controls.

We did not obtain p values in the data of genus (and species) level.

For the statistical analyses, the authors might utilize value zero (0) for the not detected elements. After the calculation, if the authors cannot obtain significant data, they should describe clearly the findings in the results section of the manuscript and alter the manuscript title.

The manuscript needs detailed description of statistical analysis in the result section.

We think that the data are very precise for the future research of BS.

2. The authors seemed to consider that the laboratory findings of next two sentences were scientifically sound.

The authors should provide direct evidence on the pro-inflammatory effects of the “bacterial and human peptides” and the “microbes of the phyla Firmicutes/Proteobacteia” in the sentences with the findings of their cited reference articles.

1) P12: The molecular mimicry based on sequence homology between microbial and human peptides triggers autoimmune responses in BS patients.

2) P12: we found Firmicutes and Proteobacteria were dominant phyla in BS patients presenting inflammatory effects in the gastrointestinal system.

Minor

1. We found several typographical errors in the manuscript.

Example

P3: Last paragraph needs the name of the disease. In the paragraph, we may recognize double space after the word “enhances”.

P8 in the next generation paragraph: Sequences wereanalyzed by ---- needs space.

P9 in the second paragraph: Groups were of similar age(p = 0.098)-----needs space.

P10 in figure legends of Fig 4: The term “genera” should be corrected.

P11 in the second paragraph: The word “fallows” should be corrected.

---

## [Author Response · Author response to Decision Letter 1]

31 Aug 2020

PONE-D-19-35822

Succinivibrionaceae is Dominant Family in Fecal Microbiota of Behçet’s Syndrome Patients with Uveitis 

PLOS ONE

Dear editor,

We revised the manuscript to address all the reviewer's comments in a point-by-point response. This rebuttal letter that responds to each point raised by the reviewers. 

Ayse Kalkanci

Corresponding author

1) Please ensure that you refer to Figure 6 in your text as, if accepted, production will need this reference to link the reader to the figure.

Figure 6 was referred in the text.

---

## [Editor Report · Decision Letter 2]

16 Sep 2020

PONE-D-19-35822R2

Succinivibrionaceae is Dominant Family in Fecal Microbiota of Behçet’s Syndrome Patients with Uveitis

PLOS ONE

Dear Dr. Kalkanci,

Thank you for submitting your manuscript to PLOS ONE. After careful consideration, we feel that it has merit but does not fully meet PLOS ONE’s publication criteria as it currently stands. Therefore, we invite you to submit a revised version of the manuscript that addresses the points raised during the review process.

We look forward to receiving your revised manuscript.

Kind regards,

Noboru Suzuki

Academic Editor

PLOS ONE

Additional Editor Comments (if provided):

We found several typographical errors in the manuscript of 2nd revision.

We strongly suggest you to consult with a Native English speaker for the correction of their English.

ie. Top 5 species in all groups were as fallow;

---

## [Author Response · Author response to Decision Letter 2]

13 Oct 2020

Response to the reviewers

"Succinivibrionaceae is Dominant Family in Fecal Microbiota of Behçet’s Syndrome Patients with Uveitis" 

1) Please amend your Response to Reviewers letter to include a point by point response to each of the points made by the Editor and / or Reviewers. Please follow this link for more information: http://blogs.PLOS.org/everyone/2011/05/10/how-to-submit-your-revised-manuscript/

Answer: Yes . “Response to reviewers-7” letter was added into submission.

2) Thank you for providing your revised Funding statement. We've made some updates to conform to journal requirements. Can you please confirm in your Cover Letter whether the following proposed statement is accurate and suitable to appear alongside your manuscript?

"This project was supported and funded with a grant of Turkish Society of Physical Medicine and Rehabilitation. The funder had no role in study design, data collection and analysis, decision to publish, or preparation of the manuscript."

If there are any errors or omissions in this statement please let us know. Otherwise, please include the above statement in your Cover Letter upon re-submission, and we will update the statement in the submission form on your behalf.

Answer: "This project was supported and funded with a grant of Turkish League Against Rheumatism. The funder had no role in study design, data collection and analysis, decision to publish, or preparation of the manuscript." This statement included in my Cover Letter.

---

## [Editor Report · Decision Letter 3]

20 Oct 2020

Succinivibrionaceae is Dominant Family in Fecal Microbiota of Behçet’s Syndrome Patients with Uveitis

PONE-D-19-35822R3

Dear Dr. Kalkanci,

We’re pleased to inform you that your manuscript has been judged scientifically suitable for publication and will be formally accepted for publication once it meets all outstanding technical requirements.

Kind regards,

Noboru Suzuki

Guest Editor

PLOS ONE
---

## [Editor Report · Acceptance letter]

22 Oct 2020

PONE-D-19-35822R3 

*Succinivibrionaceae* is Dominant Family in Fecal Microbiota of Behçet’s Syndrome Patients with Uveitis 

Dear Dr. Kalkanci:

I'm pleased to inform you that your manuscript has been deemed suitable for publication in PLOS ONE. Congratulations! Your manuscript is now with our production department. 

Kind regards, 

on behalf of

Dr. Noboru Suzuki 

Guest Editor

PLOS ONE